# Epidural Injection of Harpagoside for the Recovery of Rats with Lumbar Spinal Stenosis

**DOI:** 10.3390/cells12182281

**Published:** 2023-09-15

**Authors:** Jin Young Hong, Hyun Kim, Changhwan Yeo, Junseon Lee, Wan-Jin Jeon, Yoon Jae Lee, In-Hyuk Ha

**Affiliations:** Jaseng Spine and Joint Research Institute, Jaseng Medical Foundation, Seoul 135-896, Republic of Korea; vrt23@jaseng.org (J.Y.H.); khyeon94@jaseng.org (H.K.); duelf2@jaseng.org (C.Y.); excikind@jaseng.org (J.L.); cool2305@jaseng.org (W.-J.J.); goodsmile@jaseng.org (Y.J.L.)

**Keywords:** epidural injection, harpagoside, lumbar spinal stenosis, neuropathic pain, pain relief, inflammation

## Abstract

Epidural administration is the leading therapeutic option for the management of pain associated with lumbar spinal stenosis (LSS), which is characterized by compression of the nerve root due to narrowing of the spinal canal. Corticosteroids are effective in alleviating LSS-related pain but can lead to complications with long-term use. Recent studies have focused on identifying promising medications administered epidurally to affected spinal regions. In this study, we aimed to investigate the effectiveness of harpagoside (HAS) as an epidural medication in rats with LSS. HAS at various concentrations was effective for neuroprotection against ferrous sulfate damage and consequent promotion of axonal outgrowth in primary spinal cord neurons. When two concentrations of HAS (100 and 200 μg/kg) were administered to the rat LSS model via the epidural space once a day for 4 weeks, the inflammatory responses around the silicone block used for LSS were substantially reduced. Consistently, pain-related factors were significantly suppressed by the epidural administration of HAS. The motor functions of rats with LSS significantly improved. These findings suggest that targeted delivery of HAS directly to the affected area via epidural injection holds promise as a potential treatment option for the recovery of patients with LSS.

## 1. Introduction

Lumbar spinal stenosis (LSS) is a painful degenerative condition characterized by the narrowing of the spinal canal in the lower back, leading to nerve compression and subsequent considerable back and leg pain [1,2]. Although several nonsurgical treatment options exist, including conservative measures, such as physiotherapy, anti-inflammatory medicine, lumbosacral corset, and epidural infiltration [3,4], these methods may not always provide satisfactory pain relief or effectively address the underlying issue. However, there is a growing need for alternative therapeutic approaches owing to concerns about surgical complications [5]. Therefore, managing pain and inflammation in patients with LSS and developing innovative therapies remain significant healthcare challenges. In recent years, epidural administration has gained attention because of its direct access to the affected spinal region, enabling medication to reach the nerve roots and modulate pain relief with a minimally invasive approach in various spinal disorders [6]. The commonly used injectable medications include steroids, local anesthetics, and saline [7]. Steroids, such as dexamethasone, betamethasone, methylprednisolone, and triamcinolone, are particularly effective in alleviating pain caused by disc disease [8,9]. These corticosteroids possess potent anti-inflammatory properties and can significantly reduce inflammation around irritating nerves, thereby relieving back and leg pain and discomfort [10]. However, the long-term use of steroids in epidural injections can lead to various complications, including bleeding, infection risk, low blood pressure, headaches, nerve tissue damage, and allergic reactions [11]. In response to these concerns, the US Food and Drug Administration has issued warnings against the long-term use of epidural steroid injections and prohibited the use of particulate steroids [12]. Therefore, recent studies have focused on identifying safe and effective alternatives to provide pain relief to patients with LSS.

We first applied harpagoside (HAS) as an epidural medication in rats with LSS. HAS is a key herbal ingredient of *Harpagophytum procumbens* (HP) in Jaseng Shinbaro3 and Chungpajeon-H (Jaseng Hospital of Korean Medicine, Seoul, Republic of Korea), both of which are used clinically to treat LSS [13,14]. Previous studies have highlighted the ability of HAS to reduce the expression of inducible nitric oxide synthase (iNOS) and cyclooxygenase 2 (COX2) by inhibiting the activation of nuclear factor (NF)-kappa B in various cell types, including HepG2 and RAW 264.7 cells [15]. Specifically, focusing on its neuroprotective properties, HAS potentially attenuates dopaminergic neurodegeneration by elevating the levels of glial cell-line-derived neurotrophic factors in primary cultured mesencephalic neurons [16]. Additionally, HAS is effective in alleviating neuronal apoptosis and preventing blood–brain barrier disruption by inhibiting the toll-like receptor 4/myeloid differentiation primary response 88/NF-κB signaling pathway in an in vitro model of angiotensin II-induced BV2 microglial activation [17]. Furthermore, HAS scavenges hypoxia-enhanced nitric oxide synthesis in microglial cells derived from the primary cortex [18]. Moreover, HAS inhibits the activity of Phosphatase and TENsin (PTEN) and phosphorylated-PTEN, which are well-known inhibitors of central nervous system axon regeneration, and activates the phosphoinositide 3-kinase/Akt/glycogen synthase kinase-3 beta signaling pathway, which is necessary for axon growth in rats with chronic cerebral hypoperfusion [19]. The HP extract, which contains HAS as its primary component, improves functional recovery in LSS rat models by promoting antioxidant defense through nuclear erythroid 2-related factor 2 (NRF2)-mediated iron signal changes, and it has been mentioned that HAS may be the main cause of these effects [13].

Our study aimed to investigate the anti-inflammatory, antioxidant, and neuroprotective effects of the epidural administration of HAS and its potential for pain relief in rats with LSS. To the best of our knowledge, this is the first study to demonstrate the therapeutic effects of HAS in rats with LSS and primary cultured spinal cord neurons. Our findings suggest that the epidural administration of HAS may offer a new and promising strategy for managing LSS, suggesting its potential as a safe and effective therapeutic option.

## 2. Materials and Methods

### 2.1. Primary Rat Spinal Cord Neurons

Primary spinal cord neurons were derived from Sprague–Dawley (SD) rat embryos (Daehan Bio Link, Chungju, Republic of Korea) at 15 days of gestation. Briefly, the embryos were swiftly extracted from pregnant rats via cesarean section. They were then carefully placed in petri dishes containing pre-chilled Leibovitz’s L-15 medium (Gibco-BRL, Grand Island, NY, USA). Next, the embryonic bodies were positioned with their abdomens facing upward, and the spinal cords were meticulously isolated by removing immature spines. Under magnification, the dorsal root ganglia (DRG) and meninges were gently removed from the isolated spinal cord using fine forceps. Subsequently, the spinal cords were rinsed once in L-15 medium and enzymatically digested. The Neural Tissue Dissociation Kit and gentleMACS™ Dissociator, both supplied by Miltenyi Biotec (Bergisch Gladbach, Germany), were used for this purpose. The digestion was performed at 37 °C for 20 min. Once the dissociation was complete, the supernatants were discarded after centrifugation at 2000 rpm for 3 min. The resulting cell pellet was resuspended in 1 mL neurobasal medium supplemented with B27, GlutaMAX, 1% penicillin/streptomycin (all obtained from Gibco-BRL), and 10 ng/mL recombinant brain-derived neurotrophic factor (PeproTech, Rocky Hill, CT, USA). Subsequently, the single cells were seeded onto coated plates, which had been pretreated with a solution of 20 mg/mL poly-D-lysine overnight, followed by 10 mg/mL of laminin (both provided by Gibco-BRL) for 2 h at 4 °C.

### 2.2. Ferrous Sulfate (FeSO_4_) and Harpagoside (HAS) Treatment

After a 24 h period of cell stabilization, FeSO_4_ was pretreated in the cell culture media at a concentration of 100 μM for 30 min. Subsequently, 50, 100, or 200 μg/mL HAS was added to the cell culture medium. The cells were then subjected to a continuous incubation period of 24 h during exposure to FeSO_4_ and HAS. Following incubation, the samples were collected for subsequent analyses.

### 2.3. Neuronal Viability Assays

The Cell Counting Kit-8 assay (CCK-8, Dojindo, Kumamoto, Japan) was used to assess cell viability for the HP extract, harpagide (HAG), and HAS concentrations of 10, 25, 50, 100, 200, and 400 μg/mL, both with and without exposure to FeSO_4_. Following a 24 h incubation period, the cells in each well were treated with CCK-8 solution, constituting 10% of the total volume, for 4 h. The absorbance was measured at 450 nm using an Epoch microplate reader (BioTek, Winooski, VT, USA). Cell viability was determined by calculating the percentage of surviving neurons relative to the absorbance value of control cells, which was set at 100% viability. Furthermore, cell viability was also evaluated using a live/dead cell imaging kit (Invitrogen, Grand Island, NY, USA). The culture medium was replaced with Dulbecco’s Modified Eagle Medium (Hyclone, Logan, UT, USA) containing the dye solution, and the cells were incubated at 37 °C for 15 min. Following incubation, the cells were mounted using fluorescence mounting medium (Dako Cytomation, Carpinteria, CA, USA). Random images were captured at 100× magnification using a confocal microscope (Eclipse C2 Plus; Nikon, Minato City, Tokyo, Japan). The number of dead cells, indicated by red staining, was quantified using ImageJ (version 1.37, National Institutes of Health, Bethesda, MD, USA).

### 2.4. Immunocytochemistry

The cells were treated with 4% paraformaldehyde (PFA; Biosesang, Seongnam, Republic of Korea) for 30 min and rinsed thrice for 5 min each with phosphate-buffered saline (PBS) (Gibco-BRL). The cells were exposed to 0.2% Triton X-100 in PBS for 5 min, followed by two additional rinses with PBS. The blocking step lasted for 1 h with 2% normal goat serum in PBS. Primary antibodies, including anti-Tuj1 (1:1000; BioLegend, San Diego, CA, USA) and F-actin (1:400; Invitrogen, Waltham, MA, USA), were applied to the cells overnight at 4 °C. The cells were also incubated with rhodamine phalloidin (F-actin; Invitrogen, 1:1000) and fluorescein isothiocyanate-conjugated secondary antibodies (Jackson ImmunoResearch Laboratories, West Grove, PA, USA) for 2 h. After washing thrice for 5 min each with PBS, the cells were mounted with fluorescence mounting medium (Dako Cytomation), and images were captured at either 100× or 400× magnification, ensuring the use of consistent acquisition settings by confocal microscopy (Eclipse C2 Plus; Nikon).

### 2.5. Epidural Catheterization and the LSS Model

The experimental procedures conducted in this study strictly adhered to the guidelines and regulations of the Jaseng Animal Care and Use Committee (approval number: JSR-2023-01-005-A). Male SD rats (7 weeks old, 230–250 g) were procured from Daehan Bio Link. The rats were housed in a controlled environment at constant temperature and humidity. They had unrestricted access to food and water. To perform epidural catheterization, we followed a previously described method for epidural catheterization in rats with LSS [20]. First, the animals were anesthetized with 2–3% isoflurane gas (Forane; BK Pharm, Goyang, Republic of Korea). Subsequently, we inserted a drug delivery device and a modified catheter (Instech Laboratories, Inc., Plymouth Meeting, PA, USA) at both the C2 and T10 levels. To induce LSS after epidural catheterization in rats, silicone (length: 4 mm, width: 1 mm, thickness: 1 mm, hardness: 80 kPa) was inserted through the L4/5 space at the L4 level without performing laminectomy. The rats were divided into four groups (*n* = 20 rats/group): sham, LSS, HAS-100, and HAS-200. The sham group underwent surgery without epidural catheterization and LSS induction. The LSS group underwent surgery with epidural catheterization, epidural injection of saline solution, and LSS induction. The HAS-100 group underwent surgery with epidural catheterization, epidural injection of HAS at a concentration of 100 μg/kg, and LSS induction. The HAS-200 group underwent surgery with epidural catheterization, epidural injection of HAS at a concentration of 200 μg/kg, and LSS induction. For 4 weeks, the rats in the HAS-100 and HAS-200 groups received daily epidural injections of HAS through a drug delivery device implanted at the C2 level. The LSS group epidurally received an equivalent volume (100 μL) of saline solution. We have described the epidural catheterization and LSS induction procedures in more detail in a timetable, which we have added to the methods section as Figure 1

### 2.6. Immunohistochemistry

Immunohistochemistry was used to analyze the inflammation, pain, and axonal sprouting within the spinal cord or DRG at the silicone implantation site (L4 level). To prepare the sectioned tissues, the extracted tissues were post-fixed with 4% PFA (Biosesang) at 4 °C for 1 day. The L3–6-level spines were decalcified in a decalcification solution (BBC Biochemical, Mount Vernon, WA, USA) for 3 days, and the decalcified samples were then washed with PBS and cryoprotected in 30% sucrose for 3 days. The L4-level spine was cut at 20 μm thicknesses in the axial plane. Section were incubated with primary antibodies against rabbit anti-ectodermal dysplasia protein (ED1) (1:500; Millipore, Burlington, MA, USA), anti-iNOS (1:200; NovusBio, Littleton, CO, USA), anti-neuronal nuclear protein (NeuN) (1:500; Synaptic Systems, Göttingen, Germany), anti-transient receptor potential vanilloid type 1 channel (TRPV1) (1:100; Alomone Labs, Jerusalem, Israel), anti-neurofilament 200 (NF200) (1:200; Sigma-Aldrich, St. Louis, MO, USA), isolectin B4 (IB4) (1:100; Sigma-Aldrich), anti-calcitonin gene-related peptide (CGRP) (1:200; Sigma-Aldrich), and anti-5-hydroxytryptamine (5HT) (1:2000; Sigma-Aldrich) overnight at 4 °C. After the sections were washed thrice, secondary antibodies (FITC-conjugated goat anti-mouse or rabbit immunoglobulin G (IgG; 1:300; Jackson Immuno-Research Labs, West Grove, PA, USA), rhodamine goat anti-rabbit or guinea pig IgG (1:300; Jackson Immuno-Research Labs), Alexa Fluor 647 goat anti-mouse IgG (1:300; Jackson Immuno-Research Labs)) were incubated for 2 h at room temperature. The sections were washed thrice with PBS and mounted with a fluorescence mounting medium (DAKO). The stained tissue sections were imaged using a confocal microscope (Eclipse C2 Plus). The images for quantitative analysis were captured under 10× or 40× magnification using a confocal microscope, relative intensities were quantified using ImageJ, and the positive numbers for specific markers were manually counted.

### 2.7. Real-Time Polymerase Chain Reaction

Total RNA was isolated from the L4 spinal cord using the RNeasy Mini Kit (Qiagen, Hilden, Germany). cDNA was synthesized using oligo dT primers and AccuPower RT PreMix (Bioneer, Daejeon, Republic of Korea). Quantitative real-time polymerase chain reaction (qRT-PCR) was performed in triplicate using the iQ SYBR Green Supermix with a CFX Connect Real-Time PCR Detection System (both Bio-Rad, Hercules, CA, USA), and gene sequences are listed in Table 1. Target gene expression was normalized to that of glyceraldehyde 3-phosphate dehydrogenase and expressed as fold change relative to the LSS group.

### 2.8. Tissue Clearing

We employed a previously described tissue clearing technique to achieve transparent [21] and detailed imaging of DRG tissues. After fixing the DRG tissues with 4% PFA, we washed them with PBS. Decolorization was then performed using the CUBIC-L solution, which contains 10% Triton X-100 (Sigma-Aldrich) and 10% N-butyldiethanolamine (Tokyo Chemical Industry, Tokyo, Japan). Following decolorization, we washed the samples again with PBS to remove any remaining CUBIC-L solution. To eliminate lipids from the samples, we subjected them to dehydration using a methanol gradient. This involved immersing the samples in different concentrations of methanol (20%, 40%, 60%, 80%, and 100%) for 1 h each. Subsequently, we incubated the dehydrated samples in a solution of 66% dichloromethane (DCM, Sigma-Aldrich) and 33% methanol overnight at 4 °C to efficiently remove residual lipids. To further improve tissue clarity and prepare the samples for immunostaining, we performed delipidation by incubating the samples in a 5% hydrogen peroxide solution (Sigma-Aldrich) at 4 °C overnight. After delipidation, the samples were rehydrated using a series of methanol gradients (80%, 60%, 40%, and 20%) and PBS for 1 h each. For immunostaining, we employed a permeabilization step using a solution containing 20% dimethyl sulfoxide (DMSO, Sigma-Aldrich) and 0.3 M glycine in PTx2 (PBS with 0.2% Triton X-100). This step was carried out for 2 days at 37 °C to allow efficient permeabilization of the tissues. Following permeabilization, the samples underwent a blocking step, which involved incubation in a solution containing 10% DMSO and 6% normal goat serum (NGS) in PTx2 for 2 days at 37 °C. This blocking solution helps to reduce non-specific binding of antibodies. To label the target antigen TRPV1 and NeuN, we added a primary antibody solution of TRPV1 and NeuN diluted 1:100 and 1:500, respectively, in a solution composed of 5% DMSO and 3% NGS in PTwH (PBS with 0.2% Tween 20 and 0.1% heparin). The samples were then incubated with this antibody solution for 3 weeks at 37 °C to allow the primary antibody to bind to the target antigens. During this incubation period, we washed the samples with PTwH for 4 days to remove any unbound antibodies. After the primary antibodies’ incubation, we used a FITC or rhodamine-conjugated goat anti-rabbit or guinea pig IgG as the secondary antibodies, which were added to the samples. Following the incubation of secondary antibodies, we washed the stained samples with PTwH to remove excess secondary antibody and then dehydrated them once again in MeOH. Subsequently, the samples were immersed in a solution of 66% DCM and 33% MeOH for 3 h at room temperature. Finally, we captured images of TRPV1 and NeuN staining in cleared DRG tissues using a confocal microscope (Eclipse C2 Plus, Nikon).

### 2.9. Functional Assessments

We used four tests to assess motor function after stenosis induction and epidural HAS injection. For footprint analysis, the hindlimbs of rats were dyed with black ink, and imprinting was evaluated for three consecutive footprints on white paper (length, 50 cm; width, 13 cm). For gait parameters, stride length, step length, and toe-out angle were measured. The Basso, Beattie, and Bresnahan (BBB) scale was used, as previously described [22]. Two independent observers analyzed the hindlimb motion in an open field for 4 min. The average values were used in the analysis. The ladder walking test was used to assess the ability of the rats to maintain balance. The rats walked on a metal runway (2.5 cm between grids) from left to right thrice, and their movements were recorded using a digital camera. The score was calculated as follows: ladder score (%) = (erroneous steps of the hindlimb/total steps of the hindlimb) × 100. Locomotor function was examined in each group every 7 days for 4 weeks. The Von Frey test was used to measure the response of the rats to pain. We measured the latencies of paw withdrawal in response to mechanical stimulation applied to the center of both hind paws using a Von Frey filament (Ugo Basile, Varese, Italy). The average of three or more measurements was used. All locomotor tests were recorded using a digital camera and assessed by two observers who were blinded to the treatment conditions.

### 2.10. Statistical Analyses

All results are expressed as the means ± standard errors of the mean. Comparisons among groups were performed using one-way analysis of variance with Tukey’s post hoc analysis (GraphPad Prism, GraphPad, Inc., La Jolla, CA, USA). Differences were considered statistically significant if *p*-values were < 0.05.

## 3. Results

### 3.1. HAS Treatment Supports Axon Preservation and Growth after FeSO_4_ Injury in Primary Spinal Cord Neurons

To determine the optimal dose range for the effects of HAS on axon preservation and growth under reactive-oxygen-species-insulting conditions using FeSO_4_ in spinal cord neurons, we performed CCK and live/dead assays. In particular, the HP extract and HAG, another major component of the HP extract, were investigated. When spinal cord neurons were treated with various concentrations of HP, HAS, and HAG for 24 h, neuronal viability significantly increased at 100 and 200 μg/mL of HP, and the HAS and HAG groups showed no significant increase up to 200 μg/mL, with no cytotoxicity (Figure 1a). Following the application of HP, HAS, or HA at the same concentration range on the FeSO_4_-induced neurons, we examined neuronal viability using the CCK assay. HP and HAS have neuroprotective effects in the concentration range of 50–200 μg/mL, but HAG has no significant difference compared with the FeSO_4_ group. In addition, HAS and HAG were potentially toxic to spinal cord neurons at concentrations > 400 μg/mL (Figure 1b). Therefore, the optimal dose of HAS was at a concentration of 50–200 μg/mL, and range viability assay was further evaluated for this optimal range of HAS through imaging analysis for live and dead cells. FeSO_4_-treated neurons exhibited higher red fluorescence, whereas neurons treated with three different HAS concentrations (50, 100, or 200 μg/mL) showed a weaker red fluorescence signal (Figure 1c). The mean red fluorescence intensity showed that the HAS groups exhibited a significantly lower signal intensity in a dose-dependent manner compared with the FeSO_4_ group (Figure 1d). Confocal scanning microscopy was used to identify the axonal outgrowths stained for Tuj1 and F-actin. Axon images showed that the HAS groups had a longer diameter with richly branched axons than the smaller diameter axons in the FeSO_4_ group (Figure 1e). When measuring the axonal length from confocal images, the total and mean neurite diameters were significantly shorter in the FeSO_4_ group than those in the blank group. Compared with FeSO_4_-treated neurite length, HAS promoted a significant increase in total and mean neurite lengths at concentrations of 100 and 200 μg/mL (Figure 1f,g).

### 3.2. Epidural HAS Delivery Drives down Inflammatory Responses in Rats with LSS

We investigated the potential efficacy of epidural HAS administration in mitigating the inflammatory responses following LSS. Epidural catheterization for long-term and consecutive infusions was performed using a custom catheter and a drug injection device (Appendix A). We observed a notable abundance of ED1^+^ inflammatory cells within the tissue surrounding the implanted silicon block in the LSS group at 4 weeks (Figure 2a). However, when HAS was injected into the epidural space, a modest reduction in the infiltration of ED1^+^ cells triggered by LSS was observed. Quantitatively, there was a significant decrease in the intensity of ED1^+^ cells in the HAS group compared with that in the LSS group. However, there were no significant differences among the different HAS concentrations used (Figure 2b). iNOS, acting as an oxidant and generating peroxynitrite, plays a crucial role in initiating the inflammatory response in LSS. iNOS staining of spinal cord tissues revealed a significantly increased intensity in the LSS group, whereas the intensity was significantly reduced in the HAS group (Figure 2c). Regarding the effect difference between HAS concentrations, epidural administration of HAS at a concentration of 100 µg/mL suppressed iNOS expression more effectively than HAS-200 (Figure 2d). Furthermore, spinal cord samples were collected for the analysis of pro- and anti-inflammatory cytokines (*iNOS*, *IL-1β*, *TNF-α*, *COX2, IL-10*, and *arginase 1 [Arg1]*) with real-time PCR. We confirmed significantly elevated expression of *iNOS*, *IL-1β*, *TNF-α*, and *COX2* genes in the LSS spinal cord at 4 weeks compared with the sham spinal cord. Epidural HAS injection significantly downregulated these genes, suggesting that HAS epidural delivery can reduce the inflammatory response after LSS (Figure 2e–h). Furthermore, anti-inflammatory cytokines, such as *IL-10* and *Arg1*, were significantly upregulated by the epidural delivery of HAS in rats with LSS (Figure 2i,j). However, there was no statistically significant difference between the HAS concentrations and the dose-dependent inhibitory effect of HAS on the inflammatory response. This finding suggests that the application of HAS into the epidural space suppresses pro-inflammatory factors and promotes the activation of anti-inflammatory factors in rats with LSS.
Figure 1Survivability and axonal outgrowth of primary cultured spinal cord neurons when treated with HAS after FeSO_4_ injury. (**a**) Cell Counting Kit assay of primary spinal cord neurons treated with *HP* extract, HAS, or HAG for 24 h without FeSO_4_ injury and (**b**) with FeSO_4_ injury (*n* = 10). (**c**) Live/dead staining images for live (green) and dead (red) cells in blank, FeSO_4_-injured, and HAS (50, 100, or 200 μg/mL) + FeSO_4_-injured neurons. White scale bar = 200 µm, yellow scale bar = 50 µm. The yellow boxes indicate magnified regions in the law magnification images. (**d**) Quantitative determination of the fluorescence intensity of red-stained dead cells in each group. (**e**) Representative ICC images for Tuj1 (green) and F-actin (red) in each group. The yellow boxes indicate magnified regions in the law magnification images. (**f**,**g**) Analysis of axonal growth by measuring the mean and total neurite lengths of each group. Data are expressed as the means ± standard deviations. Significant differences were analyzed using ordinary one-way analysis of variance with Tukey’s post hoc analysis as follows: ^#^ *p* < 0.05. ^##^ *p* < 0.01, and ^####^ *p* < 0.0001 vs. blank group; * *p* < 0.05, ** *p* < 0.01, *** *p* < 0.001, and **** *p* < 0.0001 vs. FeSO_4_ group.
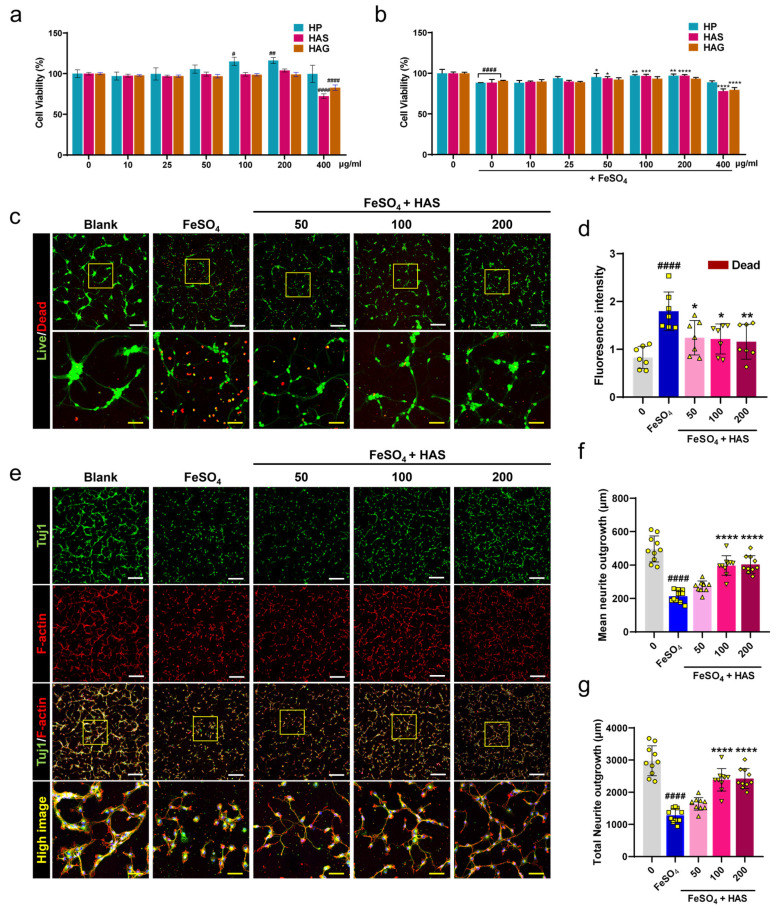



### 3.3. Epidural HAS Delivery Relieves Nociceptive Pain in DRG Neurons of Rats with LSS

To investigate the effect of epidural HAS delivery on pain arising from LSS, we examined the hyperexcitability-related expression in the DRG, which is important for pain development following nerve injury. The first marker identified in the DRG tissue was TRPV1, which is involved in the transmission of painful stimuli [23]. Immunohistochemical images showed that TRPV1 expression substantially increased in NeuN^+^ neurons in the DRG tissue after LSS, whereas epidural HAS administration markedly reduced TRPV1 expression (Figure 3a). TRPV1^+^ neurons were significantly higher in the LSS group compared with the sham group, but were significantly lower in the HAS group that received 100 or 200 μg/kg HAS epidurally (Figure 3c). Additionally, we conducted 3D imaging of TRPV1 and NeuN in sensory neurons of the cleared DRG (Figure 3b and Appendix A). Notably, the TRPV1^+^ sensory somas were found to be distributed across the entire DRG, exhibiting heightened expression following the LSS induction. Notably, we observed a significant reduction in TRPV1^+^ neurons in LSS rats that received epidural HAS injections. Concerning the expression of *TRPV1* mRNA, there was a substantial upregulation of TRPV1 gene expression, indicating a significant increase in activity after inducing LSS. However, when we administered epidural HAS, we observed a significant and dose-dependent reduction in TRPV1 gene expression (Figure 3d).

In addition, we performed triple immunostaining of DRG tissue with NF200, IB4, and CGRP. IB4-binding and CGRP receptor-containing neurons are generally involved in nociceptive pain, and NF200 is associated with proprioception. NF200 was expressed in almost all DRG neurons in each group, and it was difficult to identify differences in expression between groups; however, IB4- and CGRP-expressing neurons markedly increased after LSS. These expression rates decreased with epidural administration of HAS (Figure 4a). As a result of quantitative analysis by percentage of the number of cells expressing each marker in cell body of all neurons in the DRG tissue, there was no difference between the groups in the percentage of cells expressing NF200, but the percentage of cells expressing IB4 and CGRP was significantly increased in the LSS group and decreased in the HAS groups. However, there was no significant difference in IB4 and CGRP expression between HAS concentrations (Figure 4b–d). We also found that the expression of pain-related genes, such as *IL1RN* and *sodium voltage-gated channel alpha subunit* 9 *(SCN*9*a)*, was significantly higher in the LSS group compared with that in the sham group, whereas these genes were significantly suppressed when rats with LSS were administered HAS into the epidural space (Figure 4e,f). The observed reduction in TRPV1 expression and modulation of pain-related genes indicate that epidural HAS may serve as a promising therapeutic intervention for managing nociceptive pain arising from LSS.

### 3.4. Epidural HAS Delivery Promotes Axonal Sprouting in Rats with LSS

Axon repair has been pursued as a major treatment goal for nerve-compression-induced injuries, such as LSS. Serotonergic neurons produce the neurotransmitter serotonin (5HT) for the regrowth or sprouting of axons after nerve injury. 5HT sprouting is partly related to functional recovery, highlighting its significance in the healing process [23]. Therefore, the effect of epidural HAS injection on the promotion of 5HT^+^ axon sprouting after LSS was evaluated in the spinal cord at 4 weeks. Spinal 5HT^+^ axons sprouted after LSS induction compared with the sham group, and HAS promoted 5HT^+^ axons to sprout more robustly (Figure 5a). Further quantitative analysis revealed that the 5HT^+^ immunoreactive signals were significantly higher in the HAS group than those in the LSS group. Interestingly, the HAS-100 group demonstrated a greater spinal 5HT^+^ signal intensity than that in the HAS-200 group (Figure 5b). In particular, more active sprouting of 5HT^+^ axons was observed in the spinal dorsal horn following LSS, and this effect was augmented by epidural injection of HAS into the spinal cord (Figure 5c). To confirm the regeneration of axons that experienced compression damage during spinal canal silicone implantation, NF200 staining was performed. However, unlike the sprouting of 5HT^+^ axons, it was challenging to discern any differences between the groups in terms of NF200 axon immunosignals (Appendix A). Nevertheless, the mRNA expression of the *NF200* gene was significantly upregulated in a dose-dependent manner in the HAS group at 4 weeks (Figure 5d). These findings indicate that epidural HAS injection can effectively promote the sprouting of 5HT^+^ axons in the spinal cord following LSS, potentially contributing to the restoration of function. Although the effects on NF200 axons were not as evident, the upregulation of *NF200* expression suggests a potential underlying mechanism for axon regeneration.

### 3.5. Epidural HAS Delivery Improves LSS Recovery

Motor function was assessed weekly for up to 4 weeks after LSS using the BBB scale and ladder score, and footprint analysis was performed at the 4-week sacrifice (Figure 6a). Gait characteristics, including stride length, step length, and toe-out angle, were determined based on three consecutive hindlimb footsteps. The hindlimb gait observed in rats with LSS exhibited notable deviations compared with that in sham rats. Notably, the stride and step lengths of the hindlimbs in rats with LSS were significantly shortened, and their feet turned outward. In contrast, rats that received epidural administration of HAS exhibited stride and step lengths similar to those observed in the sham rats, indicating the restoration of normal gait patterns (Figure 6b,c). A particularly prominent characteristic observed after LSS was a significant increase in the toe-out angle. However, epidural injection of HAS in rats with LSS restored the toe-out angles to those exhibited by sham rats (Figure 6d). In addition, BBB locomotor recovery was significantly greater in the epidurally administered HAS group than that in the LSS group, starting at week 2 for all doses in the HAS group and continuing for up to 4 weeks (Figure 6e). Rats with LSS also displayed hindlimb defects when walking on a horizontal ladder, characterized by missed steps. However, epidural administration of HAS reduced the number of missed steps over 4 weeks and significantly decreased the percentage of ladder scores, indicating missed steps in the HAS-100 and HAS-200 groups at 4 weeks (Figure 6f). The latency time for the passive avoidance test was determined using the Von Frey test. Rats that received epidural injection of HAS exhibited a significantly longer latency time than those in the LSS group. Although there were no significant differences between the different concentrations of HAS, the latency time of the group that received the highest concentration (HAS 200) was similar to that of the sham group when measured weekly (Figure 6g). These findings suggest that epidural administration of HAS promotes motor recovery in rats with LSS. The restoration of gait patterns and toe-out angles, improved locomotor function, and increased latency time in the passive avoidance test indicate the positive effects of HAS on motor function.

## 4. Discussion

This study investigated the efficacy of epidural administration of HAS in improving motor function by mitigating inflammatory responses and pain in rats with LSS. HAS is a natural component found in the HP plant, also known as devil’s claw. In our previous study, we confirmed the ultimate recovery effect of the HP extract on motor function through the effective suppression of inflammation and pain relief when orally administered to rats with LSS. In particular, excessive iron deposition was observed in the spinal cord tissue damaged by LSS induction, and the HP extract inhibited the oxidative damage caused by excessive iron accumulation through its antioxidant action by activating the Nrf2 signaling pathway and maintaining iron homeostasis [13]. To maximize the effect of the HP extract on LSS treatment, we planned an efficacy evaluation study involving long-term continuous epidural administration. Therefore, we epidurally applied HAS, a major active ingredient known in the HP extract, considering the safety issues associated with epidurally administering a complex herbal formula, such as the HP extract, and the difficulty in identifying a single target.

Before administering epidural injections of HAS, we conducted in vitro evaluations of the cytotoxicity of HAS and its ability to promote axon growth. For comparison with our previous in vitro study, we used FeSO_4_ as a damaging substance for primary spinal cord neurons and measured neuronal viability by treating them with HAS, HP extract, and another major active ingredient, HAG. In the cytotoxicity evaluation, both HAS and HAG were toxic at 400 μg/mL. In contrast, the HP extract did not show significant toxicity up to a concentration of 400 μg/mL, and cell viability significantly increased at concentrations of 100 and 200 μg/mL. Under FeSO_4_ damage conditions, both the HP extract and HAS groups showed significantly higher cell viability compared with the FeSO_4_ group from concentrations of 50–200 μg/mL and significantly lower cell viability than the FeSO_4_ group at a concentration of 400 μg/mL. HAG did not show a significant increase in cell viability at a concentration of 400 μg/mL and cell viability was lower than that of the FeSO_4_ group. Therefore, we set the optimal concentration range of HAS for primary spinal cord neurons damaged by FeSO_4_ as 50–200 μg/mL, performed additional in vitro analysis to confirm the axonal-outgrowth-promoting effect, and evaluated inflammation and pain-related factors after epidural injection of HAS in rats with LSS. In the in vitro study, HAS showed significant concentration-dependent inhibition of cell death induced by FeSO_4_ and promotion of axon growth; however, in vivo, significant concentration-dependent effects were not observed. In the 5HT^+^ axonal sprouting evaluation, higher axonal sprouting was observed at the dorsal horn in the epidural administration group with an HAS concentration of 100 μg/kg compared with the group with an HAS concentration of 200 μg/kg. Epidural injection of HAS effectively regulated pain-related factors in DRG tissue. The DRG is a cluster of pseudo-unipolar neurons located in the posterior root of the spinal nerve and is involved in the occurrence and maintenance of neuropathic pain and sensory changes [24]. DRG neuron subtypes are preferentially classified according to their soma size [25]. Small-diameter DRG neurons are involved in nociceptive pain and consist of two subpopulations, CGRP^+^ peptidergic and IB4^+^ non-peptidergic nociceptive neurons, whereas large-diameter DRG neurons are classified as NF200^+^ proprioceptive neurons [26,27]. Therefore, epidural HAS injection effectively relieves nociceptive pain, including back and radicular pain, caused by LSS.

To evaluate the recovery of motor ability, a footprint test was conducted immediately before sacrifice in the 4 weeks after LSS. Paw area, paw height, and stance width were originally measured; however, there were no significant differences among the groups. However, the stride and step lengths were significantly shorter after LSS, and a frog-like gait with the soles of the feet turned outward was observed. Epidural injection of HAS significantly increased these step measurements, similar to those in the sham group, in which the soles of the feet remained parallel to the body without turning outward. Epidural administration of HAS was effective in improving motor function in LSS, including other behavioral results. However, no difference in functional recovery was observed between the different HAS concentrations.

Most other similar studies have evaluated the efficacy of one-time epidural delivery of various epidural medications such as hyaluronic acid, TNF-α inhibitor, polydeoxyribonucleotide, or biomaterials [28,29,30,31]. In addition, a polyethylene tube (PE-10) with an outer diameter of 0.61 mm was mostly used for epidural injection. Meanwhile, HAS was repeatedly injected into the epidural space for 4 weeks as in our previous paper, which standardized the long-term repeated epidural injection protocol [20]. A polyurethane tube with an outer diameter of 0.25 mm was inserted into the epidural space, which can minimize nerve damage due to epidural insertion compared to PE material [32]. Repeated long-term HAS epidural administration has been shown to have therapeutic effects in LSS rats, but the potential for toxicity increases with prolonged administration. Therefore, a limitation of our study is that further studies are needed to determine the safety of repeated epidural HAS injections in LSS rats, including factors such as body weight, food intake, organ measurements, urinalysis, hematology, blood chemistry, and histopathology.

To the best of our knowledge, this is the first study to confirm the effects of repeated long-term epidural HAS administration in the epidural space. However, no study has evaluated the effects of a single administration of HAS in rats with LSS, making it impossible to compare and evaluate repeated administrations. Additionally, our results did not clearly indicate the optimal concentration range for repeated epidural administration of HAS. Therefore, future studies are required to investigate the safety and pharmacokinetics of repeated administration of HAS and to determine the therapeutic concentration range that maximizes efficacy.

## 5. Conclusions

This study confirmed the effectiveness of the repeated long-term administration of HAS as a new epidural medication for patients with LSS. In vitro evaluations revealed that HAS treatment supports axon preservation and growth in primary spinal cord neurons, particularly under conditions of oxidative stress induced by FeSO4 injury. In vivo experiments demonstrated that the repeated epidural injection of HAS reduced inflammatory responses through decreased inflammatory cell infiltration, reduced iNOS expression, and downregulation of pro-inflammatory cytokines in LSS rats. Furthermore, it effectively regulated pain-related factors in the DRG tissue, providing relief from nociceptive pain associated with LSS. In particular, motor recovery was significantly improved along with the promotion of serotonergic (5HT^+^) axon sprouting by epidural HAS delivery. These findings offer promising insights into the potential application of HAS as a therapeutic intervention for individuals with LSS. Future studies should continue to investigate the safety, optimal dosing, and long-term effects of epidural HAS administration to advance its clinical use.

## Data Availability

The data presented in this study are available upon request from the corresponding author.

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
