# Peer review of "Epidural Injection of Harpagoside for the Recovery of Rats with Lumbar Spinal Stenosis"

_cells, 2023, doi:10.3390/cells12182281_

Round 1

Reviewer 1 Report

I commend the authors for their research entitled "Epidural Injection of Harpagoside for the Recovery of Rats with Lumbar Spinal Stenosis ". In their study the authors focused on investigation of the anti-inflammatory, antioxidant, and neuroprotective effects of the epidural administration of harpagoside (HAS), a key herbal ingredient of Harpagophytum procumbens, and its potential for pain relief in an animal model of LSS. Overall the topic is very interesting, the manuscript is well written and is scientifically sound. The discussion is based on the results. However, some points must be cleared before the manuscript could be considered for publication.
(1) Introduction. The authors stated: “HAS is a key herbal ingredient of Harpagophytum procumbens (HP) in Jaseng Shinbaro3 and Chungpajeon-H, both of which are used clinically to treat LSS.” As this plant is used in traditional Korean medicine and is not widely known in Western hemisphere, it should be introduced briefly. Please state the producer, city and country of the mentioned Jaseng Shinbaro3 and Chungpajeon-H.  
(2) Conclusions. This is the worst part of the manuscript and should be rewritten. Please expand the chapter and state clearly what is making your study unique. Describe the potential role of your results in the management of lumbar spinal stenosis, which represents the fastest growing problem in spine surgery worldwide.  
(3) References. References are scarce. I suggest to add some more to have 30+ on your list (e.g. doi:10.1007/s00586-005-0033-4).

Author Response

We wish to resubmit the manuscript titled “Epidural Injection of Harpagoside for the Recovery of Rats with Lumbar Spinal Stenosis.” The manuscript ID is cells-2574575.

We thank the editor and reviewers for their excellent and constructive comments, which clearly helped to improve the quality of this manuscript. We are pleased to provide the following point-by-point reply. Appropriate changes suggested by the reviewers have been introduced into the manuscript (highlighted within the manuscript).

We hope that our manuscript will be acceptable for publication in Cells.

Kind regards,

In-Hyuk Ha, M.D., Ph.D.

Reviewer 2 Report

Very interesting manuscript, overall well written and should contribute to the literature. Long-term use of steroids in epidural injection can lead to a variety of complications. Therefore, it is necessary to find a safe and effective alternative to relieve the pain of patients with lumbar spinal stenosis. I suggest that the author discuss further and compare the results of this study with other similar published work.

Author Response

(The authors gave the same response as above.)
